# An Outbreak of Human Systemic Anthrax, including One Case of Anthrax Meningitis, Occurred in Calabria Region (Italy): A Description of a Successful One Health Approach

**DOI:** 10.3390/life12060909

**Published:** 2022-06-17

**Authors:** Maurizio Guastalegname, Valeria Rondinone, Giuseppe Lucifora, Alfredo Vallone, Laura D’Argenio, Giovanni Petracca, Antonia Giordano, Luigina Serrecchia, Viviana Manzulli, Lorenzo Pace, Antonio Fasanella, Domenico Simone, Dora Cipolletta, Domenico Galante

**Affiliations:** 1Infectious Diseases Unit, Jazzolino Hospital, 89900 Vibo Valentia, Italy; guama@email.it (M.G.); alfredovallone@yahoo.it (A.V.); lauradi73@libero.it (L.D.); 2Istituto Zooprofilattico Sperimentale della Puglia e della Basilicata, 71121 Foggia, Italy; luigina.serrecchia@izspb.it (L.S.); viviana.manzulli@izspb.it (V.M.); lorenzo.pace@izspb.it (L.P.); antonio.fasanella@izspb.it (A.F.); domenico.simone@izspb.it (D.S.); dora.cipolletta@izspb.it (D.C.); domenico.galante@izspb.it (D.G.); 3Istituto Zooprofilattico Sperimentale del Mezzogiorno, 80055 Portici, Italy; giuseppe.lucifora@cert.izsmportici.it; 4Surgery Unit, Jazzolino Hospital, 89900 Vibo Valentia, Italy; giovanni.petracca@aspvv.it; 5Department of Prevention, Azienda Sanitaria Provinciale Vibo Valentia, 89900 Vibo Valentia, Italy; a.giordano@aspvv.it

**Keywords:** *Bacillus anthracis*, One Health, cutaneous anthrax, meningitis, antibiotic treatment, epidemiology

## Abstract

In this report, three cases of human cutaneous anthrax are described, one complicated by meningitis, and all were linked to a single infected bullock. A 41-year-old male truck driver, along with two male slaughterhouse workers, 45 and 42, were hospitalized for necrotic lesions of the arm associated with edema of the limb and high fever. All three patients were involved in transporting a bullock to the slaughterhouse. Microbiological examination on the prescapular lymph node and a piece of muscle from the bullock carcass showed the presence of *Bacillus anthracis*. The three patients underwent a biopsy of the affected tissues, and all samples tested positive for *B. anthracis* DNA using PCR. Furthermore, the truck driver also complained of an intense headache, and a CSF sampling was performed, showing him positive for *B. anthracis* by PCR, confirming the presumptive diagnosis of meningitis. Fast diagnosis and appropriate treatment are crucial for the management of human anthrax. Cooperation between human and veterinary medicine proved successful in diagnosing and resolving three human anthrax cases, confirming the reliability of the One Health approach for the surveillance of zoonoses.

## 1. Introduction

Anthrax is a zoonosis caused by *Bacillus anthracis*, a Gram-positive, aerobic, spore-forming bacillus. Animals can become infected by ingesting spores from contaminated soil or feed. Spillover to humans is usually due to direct contact with infected animals or animal products, and it can manifest as cutaneous (contact), gastrointestinal (ingestion), or respiratory (inhalation) anthrax. In addition, injectional anthrax has been reported among drug abusers, attributed to contaminated heroin [1]. All clinical forms, if untreated, could evolve into systemic anthrax. Cutaneous anthrax is the most common form described in humans [2], and in Italy, other cases have been described in the past [3,4]. About one-third of systemic disease is complicated by meningitis, characterized by a fatality rate of 92.3% [5].

In Europe, several anthrax outbreaks have recently been described among wild or domestic herbivores [6,7,8], usually occurring in late spring or summer, when high temperatures follow copious rainfalls. In Italy, anthrax is typically a sporadic disease, particularly occurring during the summer in the central and southern regions, and in the major islands, where it almost exclusively affects pastured animals [9]. Climatic conditions could affect soil suitability for anthrax, and climate change could be responsible for increased anthrax outbreaks in temperate regions in the near future [10]. Sporadic outbreaks of human anthrax have occurred in European countries during the last decades [4,11,12,13,14,15,16,17], mainly due to occupational exposure to infected animals, and very few cases of anthrax meningitis have been reported [11,12,15].

In this report, we describe 3 cases of human systemic anthrax, one of which was complicated by meningitis, all linked to a single infected bullock. The outbreak occurred near Vibo Valentia (Calabria, Italy), in a rural area where no animal anthrax outbreaks have been reported in the last decades. This case report aims to describe the interaction of human and veterinary health services engaged in the management and resolution of this human systemic anthrax outbreak and the definition of its origin.

## 2. Cases Description

On 25 May 2020, a 41-year-old man (Case 1) was admitted to “Jazzolino” Hospital in Vibo Valentia for an extensive lesion of the volar surface of the distal third of the left arm with central necrosis (Figure 1), associated with edema of the limb and high fever (40 °C). He was a truck driver who occasionally helped his father on a small family farm and reported no contact with sick or dead animals. He referred to the appearance of a small, painless itchy papule four days earlier, for which he had taken oral amoxicillin/clavulanic acid therapy (1000 mg tablets, twice a day, for 4 days) without benefit. CT scan of the limb showed widespread imbibition of the subcutaneous soft tissues reaching up to the muscle fascia with no gas collections in the tissues. Blood analysis showed neutrophilia (11.580/μL) associated with lymphopenia (540/μL) and thrombocytopenia (94.000/μL), a slight increase in muscle damage markers (creatine kinase: 576 U/L; myoglobin: 83 ng/mL) and high inflammatory markers (C-reactive protein: 238 mg/mL, procalcitonin: 0.71 ng/L).

An hour later, a 45-year-old man (Case 2) was admitted to the same hospital, presenting a centimeter lesion of the proximal third of the right forearm, covered by a central black eschar and surrounded by bullous lesions (Figure 2), associated with forearm edema and high fever (40.2 °C). He was employed in a slaughterhouse and referred to the appearance, four days earlier, of a small macular lesion, for which he had taken oral amoxicillin/clavulanic acid therapy without any benefit. Blood analysis showed neutrophilia (10.500/μL) and elevated inflammation markers (C-reactive protein: 170 mg/mL, procalcitonin: 0.42 ng/L).

Although no cases of animal anthrax have been detected in this area in the last decades, due to high clinical suspicion, a human anthrax outbreak has been suspected and reported to the local department of hygiene and public health. For both patients, blood cultures and swabs of the lesions were drawn. From these samples, cultivation (on blood sheep agar 5% for 24 h at 37 °C in aerobic condition) was performed, and no samples evidenced any bacterial growth. Considering the systemic signs, combined antibiotic treatment was set (Meropenem 1 g IV q8h + Clindamycin 900 mg q8h + Ciprofloxacin 400 mg q8h). A further investigation into the probable epidemiological link between the two patients showed that on 18 May, Case 2 went with a colleague to the farm of the father of Case 1 to take a bullock, reported as healthy by the farmer, and bring it to the slaughterhouse. Both patients were involved in the transport of the bovine. Luckily, the bullock carcass was still in cold storage in the slaughterhouse; therefore, it was easily identified and blocked for inspection before it could be placed on the market. Although the macroscopic examination of the carcass did not reveal any pathological findings, parts of the thigh muscles, prescapular lymph node, swabs from the carcass, and the blood dropped on the floor of the slaughterhouse were sent to the Istituto Zooprofilattico del Mezzogiorno, a veterinary Public Health Institute, located in Vibo Valentia, for microbiological examinations. The microscopic examination showed the presence of Gram+ bacilli (Figure 3A), subsequently identified as *B. anthracis*, by Mc Fadyean’s staining, highlighting the presence of the typical *B. anthracis* capsule staining a purplish coat around blue-stained bacilli. Also, the cultivation of the samples from the carcass and the blood dropped on the floor (on 5% blood sheep agar) showed the presence of typical non-hemolytic colonies 3–4 mm in diameter with a rough surface, similar to glass beads and with irregular margins (Figure 3B). The isolated strains, the prescapular lymph node, a piece of muscle and the bullock were sent to the Anthrax Reference Institute of Italy, located at the Istituto Zooprofilattico Sperimentale della Puglia e della Basilicata in Foggia (Apulia region), for confirmation of the diagnosis by molecular methods and for genotyping the isolated strain.

DNA was extracted from both animal samples and the *B. anthracis* isolated strain using the DNAeasy Blood and Tissue kit (Qiagen, Hilden, Germany) and following the protocols of the kit. Molecular identification of *B. anthracis* was performed using qualitative real-time PCR. The method is based on the amplification of specific three DNA sequences: *pl3* gene (part of the lambda pro-phage type 3), located on the *B. anthracis* chromosome; *cya* gene, encoding the edema factor, located on the virulence plasmid pXO1; and *capB* gene encoding the encapsulation protein, located on the virulence plasmid pXO2 [18]. The strains and all the animal samples were positive for *B. anthracis*.

At this point, a colleague from Case 2, also involved in the transportation and slaughter of the bullock, was contacted and invited to the hospital for a clinical evaluation. He was a 42-years-old man (Case 3) presenting small necrotic lesions of the distal third of the left forearm (Figure 4) with forearm edema. He reported the appearance of small, painless itchy papules on 22 May associated with high fever (39 °C), for which he had taken oral ciprofloxacin and steroid therapy with partial resolution of the systemic symptoms, but worsening of the skin lesions. After collecting the swabs of the lesion (resulting in a negative for *B. anthracis* detection), he was given a combination antibiotic treatment (Clindamycin 900 mg q8h + Ciprofloxacin 400 mg q8h). He was hospitalized for 8 days and then discharged with a prescription of oral clindamycin 600 mg q8h for 7 days and oral ciprofloxacin 500 mg q12h for 15 days and achieved complete resolution of the skin lesions.

On 26 May, the three patients underwent a biopsy of the affected tissues. The samples were sent to the Anthrax Reference Institute of Italy for microbiological analysis. All samples tested positive for *B. anthracis* DNA by PCR, but isolation of the bacterium was impossible because the patients had already been treated with antibiotics. During the day, Case 1 presented an extension of the soft tissue edema from the limb to the right hemithorax and complained of an intense headache without other signs or symptoms of meningitis. Therefore, a diagnostic lumbar puncture was performed: the chemical-physical examination of the cerebrospinal fluid (CSF) did not reveal any pathological findings, the culture was negative, and the biomolecular detection of the common CNS pathogens was also negative. Due to the suspicion of anthrax meningitis and the extensive edema, Methylprednisolone 40 mg IV twice daily was added to the treatment and then gradually tapered. Meropenem dosage was doubled. There was a rapid resolution of the systemic symptoms and a gradual improvement of the arm lesion. A CSF sample was also sent to the Anthrax Reference Laboratory and tested positive for *B. anthracis* DNA detection by PCR a few days later, confirming the presence of *B. anthracis*. The patient was discharged after 22 days of hospitalization with no systemic or neurological signs, complete resolution of the edema and marked improvement of the skin lesion. Oral doxycycline 100 mg q12h was prescribed for an additional 21 days, and the patient achieved complete resolution of the lesion.

Immediately following the institution of antibiotic treatment, Case 2 experienced rapid resolution of the systemic symptoms and a gradual improvement in the forearm lesion. Still, on day 12 of hospitalization, he presented a diffuse itchy skin rash associated with an increase in serum eosinophil count (700/μL), interpreted as a possible reaction to antibiotic treatment. Meanwhile, antimicrobial susceptibility testing, using the broth microdilution method, was performed on the *B. anthracis* strain isolated from the bullock carcass. It turned out to be sensitive to 10 different antibiotics, including penicillin, ciprofloxacin, clindamycin, and doxycycline (Table 1). Therefore, intravenous antibiotic therapy was discontinued, and oral doxycycline 100 mg q12h plus oral steroids and antihistamine drug was added to the treatment. Two days later, the rash was nearly resolved. The patient was discharged with the indication to continue treatment with oral doxycycline 100 mg q12h for a further 15 days, achieving complete resolution of the skin lesion.

AST was performed using the broth microdilution method according to Clinical and Laboratory Standards Institute (CLSI) guidelines [19,20]. The antibiotics and their tested concentrations are shown in this table. To interpret the results, the CLSI breakpoints for *B. anthracis* were used for penicillin G, amoxicillin, ciprofloxacin, levofloxacin, doxycycline, and tetracycline [19], while the breakpoints for *Staphylococcus* spp. were used for clindamycin, vancomycin, linezolid, and rifampin [20]. *Escherichia coli* ATCC 25922 and *Staphylococcus aureus* ATCC 25923 were used as control strains.

### Epidemiological Analysis

Phylogenetic analysis was determined through the research of polymorphisms for CanSNPs, with 14 PCR assays for allelic discrimination [21,22], and by core genome SNP analysis (cgSNP) [23]. SNPs were identified by mapping sequencing reads on Ames ancestor reference genome assembly. A 31-loci MLVA assay was performed to determine *B. anthracis* genotypes [9]. The analysis of 14 CanSNPs, collocated the strain into the major lineage A, sublineage A.Br.008/011, unlike a *Bacillus anthracis* strain responsible for an outbreak of anthrax that occurred in the same region in 2002 in the province of Cosenza and stored in the strain library of the Anthrax Reference Institute of Italy. The latter appears to belong to the major lineage A, sublineage A.Br.011/009, which is the most widespread in Italy. Also, the strains isolated in Umbria region in 2009 and several isolated in the Sicily region during the anthrax outbreaks that occurred in the last fifteen years belong to the sublineage A.Br.008/011. The fact that this strain belongs to a distinct lineage was also confirmed by core genome SNP analysis (Figure 5). The MLVA with 31 VNTRs analysis demonstrated a new genotype never detected in Italy before, identified as the 58th genotype, different from that previously detected in Calabria in 2002 since it showed differences in four loci (vntr 16, pXO1, vntr 19, and bams 28). The molecular analyses performed on the human samples (DNAs from cutaneous biopsies and CSF) revealed the same genetic profile of the *B. anthracis* strain isolated from the bullock, confirming the connection between human and animal infection.

## 3. Conclusions and Discussion

In conclusion, we can say that a prompt and reliable diagnosis is crucial for the management of human anthrax cases. Providing a correct and rapid anthrax diagnosis is important in addressing the correct therapy, and to avoid cutaneous anthrax cases evolving in most of the important and dangerous systemic cases, such as meningitis forms. To this aim, anthrax should always be included in the differential diagnosis of skin lesions in the presence of eschars and pustules. Cutaneous anthrax results from injecting *B. anthracis* spores through the abraded skin into subcutaneous tissues. The bacteria subsequently germinate and multiply locally and begin toxin production. This leads to the characteristic edema and cutaneous ulceration with consequent formation of black eschars. The toxin is comprised of three distinct, antigenically active components: the edema factor (EF), necessary for the edema-producing activity of the toxin, the protective antigen (PA) that induces protective antibodies, and the lethal factor (LF), essential for lethal effect. The PA is responsible for attaching the toxin to the cell, while the LF and the EF are responsible for the toxicity [29]. Accurate investigation of epidemiological and occupational aspects is needed in selected categories of workers, as well as ecological niches and environmental contaminations of the soil by *B. anthracis*, need to be considered during dry seasons preceded by rainy springs in areas where anthrax has historically occurred. Although blood cultures were negative, all patients presented criteria of systemic illness [5] which led to the prompt institution of combined antibiotic treatment. Nevertheless, the day after admission, one patient experienced severe headache, which represents an uncommon predictor sign of anthrax meningitis [5]. Therefore, a lumbar puncture was performed, and a presumptive anti-anthrax meningitis treatment was administered. CSF examination was normal, probably due to lumbar puncture being performed during the early stage of the disease [30], but *B. anthracis* DNA detection confirmed the diagnosis of meningitis. The patient completed 21 days of triple antibiotic treatment and was discharged with no systemic or neurological signs. Despite the high fatality rate of anthrax meningitis, prompt combined administration of intravenous antibiotics during the early stage of the disease led to a successful outcome, as recently reported in a case of anthrax meningoencephalitis in Romania [31]. Patients with systemic anthrax should be frequently reassessed for the appearance of signs of meningitis (including severe headache). From the epidemiological point of view, no genetic correlation has been evidenced between *B. anthracis* strains isolated in Calabria in 2020 and 2002. The strain isolated in this outbreak showed a stronger phylogenetic correlation with *B. anthracis* strains isolated in Sicily in previous anthrax outbreaks. This likely means that the entrance of this genotype in Calabria could be due to animal trade or commercial exchanges of other products (i.e., contaminated hay and feed or leather) between these two regions. The genotyping analysis with methods such as CanSNPs, cgSNP and MLVA is a very valuable tool for studying the diversity, evolution, and molecular epidemiology of *B. anthracis*, and for understanding the dynamic of diffusion of this bacterium as well as the eventual correlations between animal and human cases.

Furthermore, this case report highlights the importance of veterinary services in the prevention of anthrax since an effective system of surveillance is fundamental to tracking the disease and stopping its spread. If we enhance the surveillance systems, we can also reduce the risk of transmission to humans, as was observed in this case. To this aim, we also believe that animal vaccination is an extremely important instrument to prevent anthrax and avoid its diffusion, especially in those areas where anthrax is endemic.

Another important aspect emerging from this episode is the role of subclinical forms of anthrax in animals [32] since the slaughtered bullock, as referred by the farmer, didn’t show evident clinical signs of anthrax when alive. Based on this, a subclinical form cannot be excluded. This aspect needs further investigations, such as serological tests on the animals living in the area where this anthrax outbreak occurred.

The occurrence of animal and human anthrax cases highlights the importance of the concept of One Health, with prompt cooperation between clinicians and veterinarians, the healthcare delivery system and public health officials. Even though human anthrax is very rare in Italy, this interdisciplinary approach was once again successful and decisive, as happened for two past anthrax outbreaks involving three people affected by cutaneous anthrax, one case in 2012 in the Basilicata region [3], and two cases in 2017 in the Lazio region [4]. The affected people in those cases were two farmworkers and one veterinary. Coordination among veterinary services, local and referral hospitals, epidemiology services, and research institutions allowed the identification of three cases of human cutaneous anthrax associated with this outbreak. It confirmed the reliability of the One Health approach for surveillance of zoonoses. Further steps are needed to strengthen the epidemiological, clinical, and diagnostic competence regarding old diseases remerging today. This case represents a perfect example of cooperation between human and veterinary medicine in diagnosing and correctly solving three human anthrax cases.

## Figures and Tables

**Figure 1 life-12-00909-f001:**
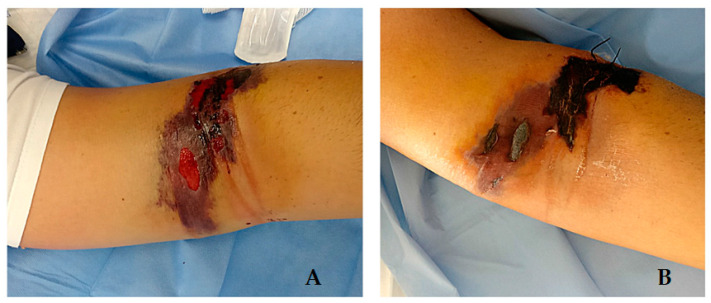
Anthrax skin lesion Case 1: (**A**) 8 days after exposure; (**B**) 15 days after exposure.

**Figure 2 life-12-00909-f002:**
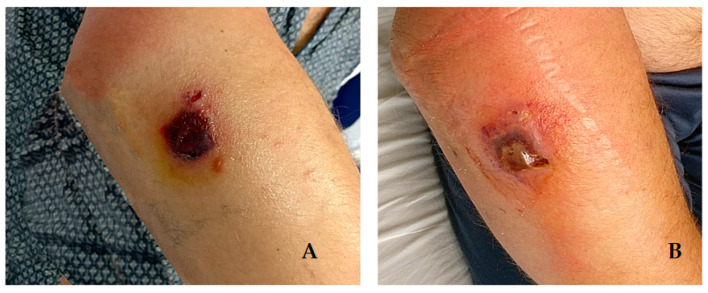
Anthrax skin lesion Case 2: (**A**) 8 days after exposure; (**B**) 20 days after exposure.

**Figure 3 life-12-00909-f003:**
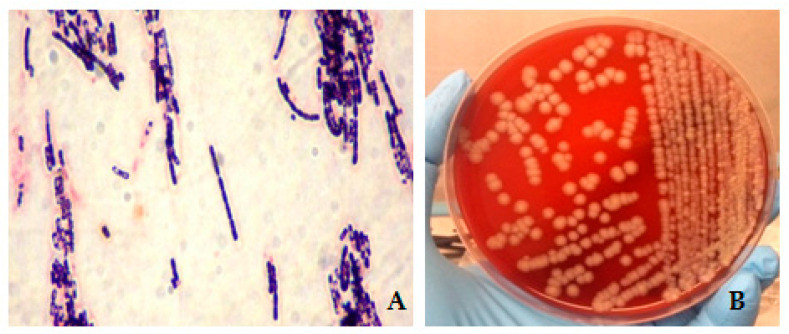
(**A**) Gram stain showing elongated Gram + bacillary forms. (**B**) *B. anthracis* typical colonies after cultivation on blood sheep agar 5%, isolated from the bullock carcass and the blood dropped on the slaughterhouse floor.

**Figure 4 life-12-00909-f004:**
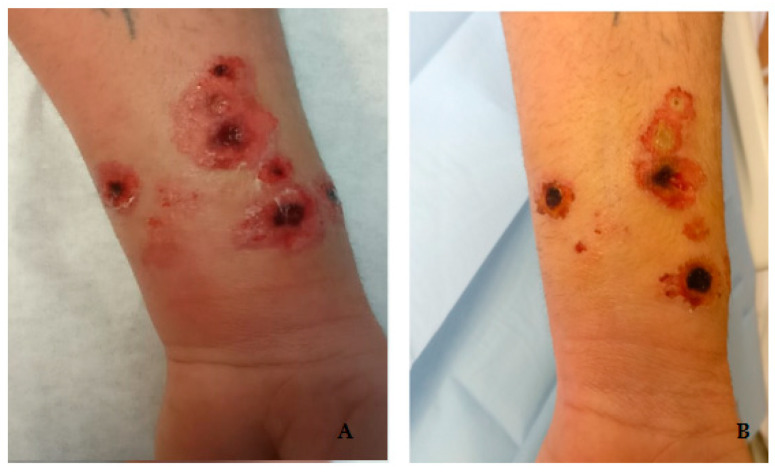
Anthrax skin lesions Case 3: (**A**) 8 days after exposure; (**B**) 15 days after exposure.

**Figure 5 life-12-00909-f005:**
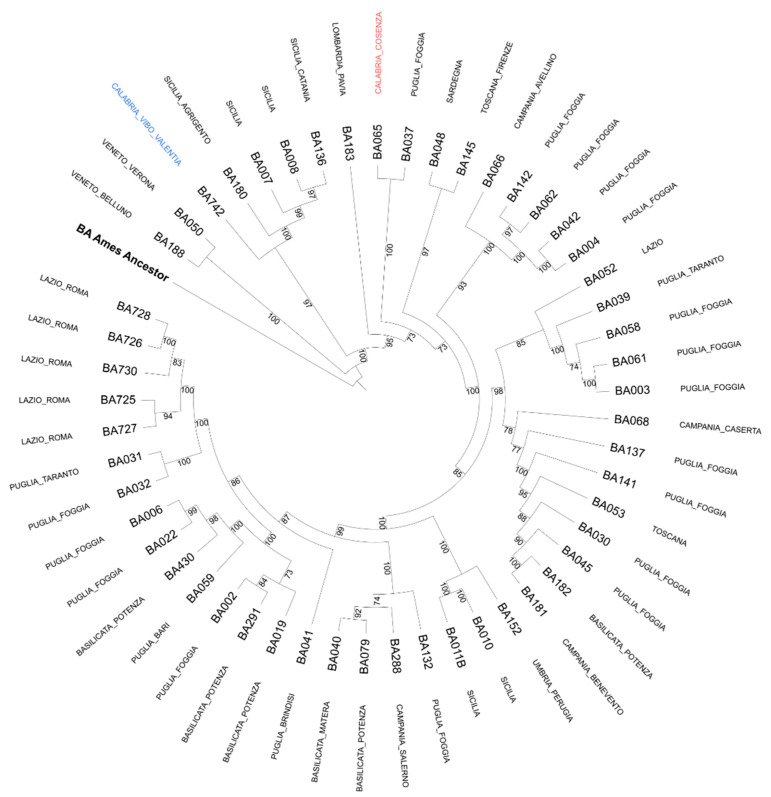
**Phylogenetic tree of *B. anthracis* isolates, including the strain (BA742) surveyed in this study.** Draft genomes were assembled from raw data with SPAdes (v3.15) [24] and annotated with PROKKA (v1.14.6) [25]. The core genome used for tree inference was calculated with Roary (v3.13) [26]. The core genome SNP (cgSNP) tree was calculated with FastTree (v2.1) [27] using the GTR (generalized time-reversible) model and decorated with iTOL (v6) [28]. *Bacillus anthracis* strain Ames (in bold) was used as the type strain to root the tree. It can be noticed that there is no strict correlation between the strain isolated in this study (blue color) and another strain isolated in the same region (Calabria region) in 2002 (red color). The strain BA742 seems to be more similar to other strains isolated in Sicily during outbreaks that occurred in the past years.

**Table 1 life-12-00909-t001:** Antimicrobial susceptibility testing on *B. anthracis* strain isolated from the bullock carcass.

Antimicrobial Agent	MIC Range (µg/mL)	MIC Breakpoints ^a^ (µg/mL)			MIC Value (µg/mL)		
		S (≤)	I	R (≥)	S	I	R
Penicillin G	0.001–32	0.5	-	1	0.25		
Amoxicillin	0.004–0.5	0.12	-	0.25	0.125		
Clindamycin	0.031–4	0.5	1–2	4	0.25		
Vancomycin	0.25–32	2	4–8	16	2		
Linezolid	0.125–8	4	-	8	1		
Tetracycline	0.001–2	1	-	-	0.0625		
Ciprofloxacin	0.008–1	0.25	-	-	0.125		
Levofloxacin	0.008–1	0.25			0.125		
Doxycycline	0.001–2	1	-	-	0.031		
Rifampin	0.031–4	1	2	4	0.5		

^a^ S: susceptible; I: intermediate; R: resistant.

## Data Availability

The data presented in this study are available on request from the corresponding author.

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
