# Peer review of "An Outbreak of Human Systemic Anthrax, including One Case of Anthrax Meningitis, Occurred in Calabria Region (Italy): A Description of a Successful One Health Approach"

_life, 2022, doi:10.3390/life12060909_

Round 1
Reviewer 1 Report
The case report “An outbreak of human systemic anthrax, including one case of anthrax meningitis, occurred in Calabria region (Italy): a description of a successful One Health approach” describes three cases of human cutaneous anthrax, one of which was complicated by meningitis, and all linked to a single infected bullock. In my opinion, this is an interesting report and the authors presented the case in a good manner. I have a few minor comments the authors may consider to improve the quality of this case report.
1. In the introduction, the authors discussed three different cases of anthrax, I would suggest incorporating the aim of this case report in the introduction part as well.
2. Could the authors be more specific on the empirical therapy “amoxicillin/clavulanic acid therapy without benefit”?
3. Can authors be precise on the techniques used to isolate bacteria for instance “For both patients, blood cultures and swabs of the lesions were drawn (culture performed on all samples were negative)”?
4. Please do discuss the pathophysiological mechanism of the development of edema formation and black eschar formation in cutaneous anthrax.
Author Response
REVIEWER 1
The case report “An outbreak of human systemic anthrax, including one case of anthrax meningitis, occurred in Calabria region (Italy): a description of a successful One Health approach” describes three cases of human cutaneous anthrax, one of which was complicated by meningitis, and all linked to a single infected bullock. In my opinion, this is an interesting report and the authors presented the case in a good manner. I have a few minor comments the authors may consider to improve the quality of this case report.
- In the introduction, the authors discussed three different cases of anthrax, I would suggest incorporating the aim of this case report in the introduction part as well.
Thanks for your comment. We added the aim of this case report also in the introduction part.
- Could the authors be more specific on the empirical therapy “amoxicillin/clavulanic acid therapy without benefit”?
Thanks for your comment. We added the posology and the duration of the therapy in the manuscript.
- Can authors be precise on the techniques used to isolate bacteria for instance “For both patients, blood cultures and swabs of the lesions were drawn (culture performed on all samples were negative)”?
Thanks for your comment. We added this sentence to the manuscript “From these samples cultivation (on blood sheep agar 5% for 24 hours at 37°C in aerobic condition) was performed and for all the samples was not evidenced any bacterial growth”.
- Please do discuss the pathophysiological mechanism of the development of edema formation and black eschar formation in cutaneous anthrax.
Thanks for your comment. We added a short description of the pathophysiological mechanism of the development of edema formation and black eschar formation in cutaneous anthrax in the “conclusions and discussion section”. We cannot dwell too much on this aspect since we have limits of words for this kind of manuscript.
Reviewer 2 Report
The case report “An outbreak of human systemic anthrax, including one case of anthrax meningitis, occurred in Calabria region (Italy): a description of a successful One Health approach" shows interesting results about three cases of human cutaneous anthrax, one of which complicated by meningitis, and all linked to a single infected bullock that researchers in this field may be interested to take a look on. However, the manuscript has some faults that should be checked before publication.
Page 3. Line 111. The authors said that the microscopic examination showed the presence of gram+ bacilli. It is not clear to this reviewer the origin of these bacteria. Did they were isolate from the carcass or the blood dropped on the floor of the slaughterhouse? Pls clarify.
I believe that figures 3 and 4 could be merged into one (Fig. 3 A and B). Again, were these images of bacteria isolated from the carcass or the blood dropped on the floor of the slaughterhouse?
Page 4. Line 133. How were the DNA templates for PCR purified from the strains and all animal samples?
Fig 5A should be replaced for another of better resolution.
Author Response
REVIEWER 2
The case report “An outbreak of human systemic anthrax, including one case of anthrax meningitis, occurred in Calabria region (Italy): a description of a successful One Health approach" shows interesting results about three cases of human cutaneous anthrax, one of which complicated by meningitis, and all linked to a single infected bullock that researchers in this field may be interested to take a look on. However, the manuscript has some faults that should be checked before publication.
- Page 3. Line 111. The authors said that the microscopic examination showed the presence of gram+ bacilli. It is not clear to this reviewer the origin of these bacteria. Did they were isolate from the carcass or the blood dropped on the floor of the slaughterhouse? Pls clarify.
Thank you for your comment, B. anthracis was isolated both from swabs on the carcass and from the blood dropped on the floor of the slaughterhouse. This point was clarified also in the manuscript.
- I believe that figures 3 and 4 could be merged into one (Fig. 3 A and B). Again, were these images of bacteria isolated from the carcass or the blood dropped on the floor of the slaughterhouse?
Figures 3 and 4 were merged. Bacteria were isolated both from swabs on the carcass and from the blood dropped on the floor of the slaughterhouse. This point was clarified also in the manuscript.
- Page 4. Line 133. How were the DNA templates for PCR purified from the strains and all animal samples?
Thank you for your comment. DNA was extracted from both animal samples and B. anthracis isolated strain using the DNAeasy Blood and Tissue kit (Qiagen, Hilden, Germany). This sentence was added to the manuscript.
- Fig 5A should be replaced for another of better resolution.
Thank you for your indication. The figure 5A was replaced with another one of better resolution.
Reviewer 3 Report
The authors have raised an interesting topic regarding the anthrax meningitis, that occurred in the Calabria region (Italy). It provides insight into a successful One Health approach, which might be an exciting topic to the readership.
However, after I read through the content of this article, I found the sample group might be too little for the proof of the ideas, which is only three related cases. This won't contribute to the soundness of scientific research. hence i would suggest whether more samples should be included in the analysis.
Moreover, as the methodology proof of the cases, I would suggest more quantitative analysis shall be taken into the consideration, include the RNA or DNA order or disorder, or any other bio-markers which can show more connection between different cases. I would like to see more of this portion in your next improved version.
Author Response
Dear reviewer, thank you very much for your valuable comments. Anthrax in Italy is a sporadic disease and human cases are really rare. Furthermore, this case report also showed the particularity of one human case of anthrax meningitis, that to our knowledge was never described before in Italy. The aim of our paper was just to describe the approach used in this anthrax outbreak, in particular the cooperation between human and veterinary health services engaged in the management, resolution, and definition of the origin of this anthrax outbreak.
Based on the above, unluckily, it’s not possible to expand the sample group, since all the people and the animals involved in this anthrax outbreak have been already considered and described in this manuscript.
As regards the methodology, we could not discuss too much about the molecular analyses and the related information, because we chose the format of the case report, and the article could not be too much long, so, we would have taken us way over the number of possible words. We described in a synthetic, but exhaustive way all the analyses (PCR, CanSNPs, cgSNPs, MLVA 31 loci) necessary to prove that there was connection between the human and animal cases.
Round 2
Reviewer 2 Report
The authors have satisfactorily responded to all my questions and made the necessary changes to the manuscript.
Author Response
Dear Reviewer thank you very much for helping us to improve our manuscript.
Reviewer 3 Report
This version gets improved in the method and result discussion.
however, there is still lack of evidence or group size for the proof, hence I would suggest to put more before the acceptance.
Author Response
This version gets improved in the method and result discussion. However, there is still lack of evidence or group size for the proof, hence I would suggest to put more before the acceptance.
Dear Reviewer,
as we told in the previous review round, this is a descriptive case report, and we already described all the cases of this anthrax outbreak. As regards this case report we cannot add other cases, there are not more. However, to give more strenght and proof of the fruitful collaboration between human and veterinary medicine services, as you asked, we added in the discussion section a short summary of the last cutaneous anthrax outbreaks (with 3 more human cases) occurred in Italy in the last years (with the relative bibliographic references) and in which we were involved as veterinary services in order to provide our support.